# Oxidation during Fresh Plant Processing: A Race against Time

Didier Barmaverain, Samuel Hasler , Christoph Kalbermatten *, Matthias Plath and Roger Kalbermatten

Department of Research and Development, Ceres Heilmittel AG, 8593 Kesswil, Switzerland;
didier.barmaverain@ceresheilmittel.ch (D.B.); samuel.hasler@ceresheilmittel.ch (S.H.);
matthias.plath@ceresheilmittel.ch (M.P.); roger.kalbermatten@ceresheilmittel.ch (R.K.)
* Correspondence: research@ceresheilmittel.ch; Tel.: +41-71-466-82-81

**Abstract:** Oxidation is a major concern in the food and cosmetics industry; however, little information is available in the literature about its effect during the production of herbal medicines. The impact of oxidation on herbal mother tincture (MT) manufacturing was investigated by performing an oxidative stress test, in which cryogenically ground fresh plants (*Echinacea purpurea*, *Mentha piperita*, *Ginkgo biloba*, and *Hypericum perforatum*) were exposed to air in a time-controlled manner before extraction. The effect of oxidation on the resulting extracts was evaluated using UV–Vis spectroscopy and potassium permanganate antioxidant assay. Furthermore, a tyrosinase enzymatic assay was performed on MTs to evaluate the behavior of the absorbance spectra of phenolic compounds during oxidation. Additionally, several commercially available herbal mother tinctures were examined for oxidative changes. The exposure of the fresh plant material to air for 30 min decreased the antioxidant activity in all four tested plants by 10% to 44%. This decrease occurred along with an intensity diminution and flattening of the typical UV–Vis absorption spectra of the MTs. The results have shown that the impact of oxidation during MT manufacturing is a serious issue and could be monitored by means of simple UV–Vis spectra recording.

**Keywords:** herbal mother tinctures; oxidation; polyphenol oxidase; UV–Vis spectrophotometry; antioxidant assay



## 1. Introduction

The role of oxidation in the food and cosmetic industries, with regard to the quality of final products and their stability over time, is a well documented area of study in the literature [1–3]. Countless investigations into plant polyphenols and the deleterious effect of their oxidation on the final quality of processed products have been carried out, including studies on the browning problem in the food industry [4–6]. However, there has not been enough research conducted with the aim of evaluating the impact of oxidation during the manufacture of herbal remedies in phytopharmaceutical companies. During the preparation of herbal mother tinctures (MTs) in particular, where plants are processed fresh, being able to ensure the good preservation and conservation of the active components of the plants is of vital importance for the characteristics of the final remedy [4]. Currently, quality control analysis seems to lack methods to assess the possible oxidative impact of manufacturing processes on MTs. Phenolic compounds (PCs) are some of the most oxidation-sensitive active plant metabolites and their content can be affected by exposure to air and excessive heating during the cutting and grinding process in common industrial blenders [4,5]. One of the main contributors to the oxidative reaction is the enzyme tyrosinase, also known in plants as polyphenol oxidase (PPO). This enzyme is a multicopper glycoprotein that is capable of reacting with phenolic compounds in the presence of oxygen as soon as the integrity of the plant cell is compromised during the size reduction process. Once the cell is disrupted, the phenols stored in the vacuole can interact with the PPO stored in the plastids and catalyze the oxidation of the hydroxyl groups typical of PCs (Figure 1a) [6]. The resulting oxidized products, called benzoquinones,

are unstable species that undergo further reactions with nucleophilic substances such as phenols, amino acids, or proteins, producing complex polymers, known as melanins, which are responsible for the browning effect [7]. Phenolic compounds are well known for their ability to exert an important antioxidant radical scavenging activity in the human body. Recent studies have shown that the free radical scavenging capacity is enabled by the presence of hydroxyl groups linked to the benzene ring of PCs. This structure is able to donate hydrogen free radicals (H) and capture and stabilize free radical species by the delocalization of the unpaired electron within the aromatic conjugated system (Figure 1b) [8]. Phenolic compounds are distributed widely in the plant kingdom and their antioxidant activity plays an important role in the biological activity of several MTs. If phenolics undergo oxidation during the processing of fresh plants, the loss of the hydroxyl group (as shown in Figure 1a) should result in a decrease in their antioxidant activity, as this structure is crucial for the radical scavenging effect.

**Figure 1.** Oxidation of *orto*-diphenol in *orto*-quinone catalyzed by polyphenol oxidase (PPO) in the presence of oxygen and subsequent transformation in melanins (**a**). Suggested mechanism of free radical scavenging effect of *o*-diphenol against free radical species. X = structure of generic phenolic compound (**b**).

The aim of this research is to use a simple analytical method that can be included in a routine assay for the rapid assessment of the oxidative status of an MT. To do this, we chose to simulate an oxidative process during the processing of fresh plants and evaluate its impact on the final MT by means of antioxidant assay and UV–Vis spectra recording. Several methods can be used in vitro to assess the antioxidant activity of plant extracts; among these, one of the most well known in the literature is the use of the DPPH free radical scavenging assay [9]. Due to its availability in our laboratory, lower toxicity, and similar tested efficacy in assessing free radical scavenging activity, we chose to perform the antioxidant test using potassium permanganate ($KMnO_4$) instead of DPPH [10]. Furthermore, the promising results found in the literature for the UV–Vis spectra investigation of phenolic compound oxidation based on the use of mushroom tyrosinase [11–13] encouraged us to include this additional enzymatic oxidation test in our experimental workflow.

## 2. Materials and Methods

### 2.1. Chemicals

Potassium permanganate ($\geq$99%) was purchased from Sigma-Aldrich (Darmstadt, Germany). Potassium hydroxide (extra pure), potassium dihydrogen phosphate ($\geq$99.5%), and dipotassium hydrogen phosphate ($\geq$99.5%) were purchased from Scharlau (Barcelona, Spain). Distilled water was prepared using a Millipore Q water purification system (Millipore, Billerica, Massachusetts). Liquid nitrogen was purchased from Pangas (Dagmersellen, Switzerland) and mushroom tyrosinase (>1000 Ui) was obtained from Sigma-Aldrich (Darmstadt, Germany).

## 2.2. Herbal Material

The fresh aerial parts of *Echinacea purpurea* (L.) Moench (flowers, stems, and leaves) and *Mentha piperita* (L.) (stems and leaves) used for the preparation of the MTs in the oxidation stress test were obtained from Ekkharthof (Lengwil, Switzerland). Fresh leaves of *Ginkgo biloba* (L.) were harvested in Uttwil (Switzerland). The fresh aerial parts of *Hypericum perforatum* (L.) were harvested at the beginning of flowering in Heiden (Switzerland).

## 2.3. MTs Samples

Samples of *Echinacea purpurea*, *Mentha piperita*, *Salvia officinalis*, *Ginkgo biloba*, and *Hypericum perforatum* MTs were obtained from Ceres Heilmittel AG (Kesswil, Switzerland) and local pharmacies and stored at room temperature ($23 \pm 2$ °C). The different tinctures were conventionally named with a code consisting of a capital letter "C" for tinctures from Ceres Heilmittel AG and preceded by a number (n) for mother tinctures obtained from local pharmacies. The letter was followed by two numbers indicating the year of production (e.g., "C18g", "1MT18g", "2MT18g", and"3MT18g" indicate *Ginkgo* mother tinctures produced in 2018 by Ceres and other manufacturers; each of the digits 1–3 denotes a different manufacturer). An additional lowercase letter is added at the end of the code to distinguish which plant the mother tincture refers to ("e" = *Echinacea purpurea*; "g" = *Ginkgo* biloba; "h" = *Hypericum perforatum*; "p" = *Mentha piperita*; "s" = *Salvia officinalis*). All MTs were prepared according to the German Homeopathic Pharmacopeia (GHP) method 3a [14] (except for *Echinacea* MT coded as 5MTe, where the formulation also contains plant roots).

## 2.4. Standard GHP Method for MT Control (Dry Residue, Relative Density)

The dry residue of each MT sample was determined using the Moisture Analyzer DBS 60-3 (Kern). First, 2 g of undiluted MT was placed in DBS 60-3 and evaporated at 105 °C. The remaining weight was recorded and expressed as the percentage of drying loss. The relative density of each MT was evaluated by submitting 2 mL for analysis using the densitometer DMA 4100 M (Anton Paar). Both methods were performed following the directions of the GHP [15].

## 2.5. Extract Preparation (Oxidation Stress Test)

The fresh aerial parts of the selected medicinal plants were processed immediately after harvesting by adapting the 3a method described for the GHP. The plants were cut and cryo-ground manually for 2.5 min with liquid nitrogen. The cryogenic grinding procedure was selected due to the capacity of nitrogen to guarantee an inert environment as it displaces oxygen and also reduces enzymatic processes to a minimum, due to its low temperature [16]. The reference extract "oxidation free" (CRYO) was prepared by adding the hydro-alcoholic solution to the ground material immediately after grinding. A second extract "oxidized sample" (CRYO_30) was instead prepared by adding the solvent to the ground material only after 30 min, during which the ground plant was left in contact with air to oxidize. For each extract, 15 g of ground material was weighted after grinding and macerated with ethanol 86% ($w/w$) in a 50 mL falcon plastic tube, where it was protected from light under an airtight seal. The amount of ethanol 86% ($m/m$) to be added was calculated using Equation (1), with $m$ being the mass of the fresh herbal material and $T$ being the water percentage loss after the drying of the starting material.

$$\text{Ethanol } 86\% \ (w/w) = (2 \times m \times T)/100 \qquad (1)$$

After 15 days at room temperature, the macerated samples were centrifuged at 4000 rpm for 20 min (Heraeus Megafuge) and the upper phase was recovered and filtered through a 12–25 µm filter (Black Ribbon) into a 20 mL volumetric flask. The oxidation stress test was performed on plant samples recovered from the Ceres Heilmittel AG production department (Kesswil, Switzerland) in the summer 2021 (*Echinacea purpurea*, *Ginkgo biloba*, *Hypericum perforatum* and *Mentha piperita*). *Salvia officinalis* was transformed in Nax,

Switzerland, instead of Kesswil, and for this reason was not included in the oxidation stress test.

### 2.6. UV–Visible Spectrophotometry

Spectroscopic analysis was conducted using a Specord 200 plus spectrophotometer Analytik Jena (Thuringia, Germany) and 3 mL quartz cuvette. Absorption spectra of 1:250 aqueous dilutions of the MTs were recorded in the range of 200–500 nm. Measurements and the recording of values of maximum and minimum peaks were performed using the AspectUV software 2020 (version 1.4.4), Analytik Jena AG, Jena, Germany. The raw data were exported and processed in Microsoft Excel 2022 (version 16.62), Microsoft, Redmond, WA, USA.

### 2.7. Tyrosinase Oxidation Assay

The tyrosinase oxidation assay was conducted by adapting the enzymatic assay described by Tono et al. 1987 [11] and Oda et al. 1989 [17], with few modifications. In a 3 mL quartz cuvette, 2 mL of 50 mM phosphate buffer solution (pH 6.8) was mixed with 0.25 mL of the MT (diluted 1:25 *v/v* with the previous buffer) and 0.25 mL of the tyrosinase solution (8.3 U/mL for *Echinacea*, *Salvia* and *Mentha* and 16.2 U/mL for *Hypericum* and *Ginkgo*). Repeated scans (recorded every minute) of the UV−Vis absorption spectra over the 200−500 nm range were taken over 60 min and subsequently at the time points of 90 min, 120 min, and 180 min. The reaction sample was measured against a blank containing all the same reagents except for a 60% *w/w* ethanol dilution (1:25 *v/v*), which was used instead of the MT dilution. The first spectrum was recorded immediately after the addition of the tyrosinase solution.

### 2.8. Potassium Permanganate (KMnO$_4$) Antioxidant Assay

Antioxidant activity, expressed as the gallic acid equivalent (GAE), was measured by adapting the potassium permanganate radical scavenging assay described by Amponsah et al. 2016 [10]. A potassium dihydrogen orthophosphate (KH$_2$PO$_4$) buffer solution was prepared by dissolving 8.7 g of KH$_2$PO$_4$ in 400 mL of distilled water and adjusting it with 1 M potassium hydroxide (KOH) to pH 9. To conduct the test, 3 mL of KMnO$_4$ (80 mg/L) was added to 1 mL of a 1:250 *v/v* aqueous dilution of the MT sample. After 30 min at $22 \pm 2$ °C, in the absence of light, absorbance was measured against the buffer solution at 525 nm. A sample containing 1 mL of distilled water + 3 mL of the KMnO$_4$ solution was used as a blank. Distilled water was used to prepare 1:250 dilutions of MT to avoid ethanol interference reacting with the KMnO$_4$. The scavenging effect was expressed as GAE using the calibration curve ($r^2 = 0.995$; $y = 1.3955x + 1.7649$) prepared with gallic acid in the range of 0–48 µg/mL. Results were corrected, respectively, to the blank solvent and normalized on a dry residue. Each test was conducted in triplicate and the results were expressed as GAE per dry residue $\pm$ standard deviation (mg/mL).

### 2.9. Statistical Analysis

Significant differences in the MTs analyzed using the KMnO$_4$ scavenging activity were highlighted using a one-way analysis of variance, followed by a Tukey HSD pairwise test using R programming (version 4.1.0), Microsoft, Redmond, WA, USA, for statistical analysis. These tests are often used to compare significant differences ($p$-value $\leq 0.05$) between multiple data groups. The respective quantitative results were expressed as the mean $\pm$ standard deviation of at least three replicates for each experiment and calculated using Microsoft Office Excel 2022 (version 16.62), Microsoft, Redmond, WA, USA.

## 3. Results

### 3.1. Standard GHP Method for MTs Control (Dry Residue, Relative Density)

The dry residue percentage of the analyzed MTs, representing the total amount of extracted substances and the respective relative density, is represented in Table 1. All investigated MTs were in accordance with the specifications of the GHP.

**Table 1.** Results of MTs laboratory analysis.

| Analyzed Samples | GAE Dry Res. Normalized (mg/mL) (KMnO$_4$ Assay) | Max [a]/Min [b] Ratio (UV–Vis Spectra) | * Dry Residue | ** Rel. Density |
|---|---|---|---|---|
| *Echinacea purpurea* | | | | |
| CRYOe | 7.67 ± 0.68 | 2.66 | 2.67% | 0.9072 |
| C16e | 6.70 ± 0.07 | 1.71 | 2.05% | 0.9056 |
| C18e | 5.63 ± 0.27 | 1.67 | 2.78% | 0.9086 |
| C19e | 5.47 ± 0.29 | 1.70 | 2.45% | 0.9083 |
| 4MT16e | 5.14 ± 0.33 | 1.67 | 2.4% | 0.9128 |
| 5MTe | 4.96 ± 0.49 | 1.37 | 1.69% | 0.9065 |
| CRYO_30e | 4.26 ± 0.05 | 1.79 | 1.98% | 0.9055 |
| 1MT18e | 4.00 ± 0.44 | 1.18 | 1.89% | 0.9049 |
| 3MT18e | 3.36 ± 0.03 | nd | 2.31% | 0.9077 |
| 2MT19e | 2.48 ± 0.16 | 0.90 | 2.41% | 0.9046 |
| *Salvia officinalis* | | | | |
| C18s | 11.27 ± 0.13 | 1.18 | 3.13% | 0.9073 |
| C17s | 11.19 ± 0.22 | 1.09 | 3.50% | 0.9070 |
| 4MT18s | 9.26 ± 0.20 | 1.04 | 3.36% | 0.9111 |
| 2MT17s | 7.91 ± 0.14 | 0.83 | 2.65% | 0.8985 |
| 1MT17s | 7.79 ± 0.16 | 0.83 | 3.67% | 0.9035 |
| *Hypericum perforatum* | | | | |
| CRYOh | 11.66 ± 0.34 | 1.04 | 3.13% | 0.9072 |
| C17h | 11.12 ± 0.46 | 1.02 | 3.50% | 0.9109 |
| C18h | 10.53 ± 0.09 | 1.03 | 3.36% | 0.9115 |
| CRYO_30h | 9.37 ± 0.17 | nd | 2.65% | 0.9057 |
| C19h | 9.01 ± 0.38 | 1.01 | 3.67% | 0.9118 |
| 2MT16h | 7.95 ± 0.32 | nd | 2.42% | 0.9102 |
| 1MT16h | 7.70 ± 0.08 | nd | 2.39% | 0.9102 |
| 3MT18h | 6.74 ± 0.20 | nd | 3.17% | 0.9089 |
| C17h | 11.12 ± 0.46 | 1.02 | 3.50% | 0.9109 |
| C18h | 10.53 ± 0.09 | 1.03 | 3.36% | 0.9115 |
| CRYO_30h | 9.37 ± 0.17 | nd | 2.65% | 0.9057 |
| *Ginkgo biloba* | | | | |
| C18g | 6.28 ± 0.14 | nd | 6.91% | 0.9228 |
| 4MT15g | 6.16 ± 0.11 | nd | 4.3% | 0.9095 |
| C19g | 5.94 ± 0.18 | nd | 5.39% | 0.9162 |
| 3MT18g | 5.80 ± 0.29 | nd | 5.16% | 0.9183 |
| CRYOg | 5.59 ± 0.08 | nd | 6.02% | 0.9175 |
| C17g | 5.40 ± 0.22 | nd | 5.34% | 0.9196 |
| 5MTg | 5.38 ± 0.27 | nd | 4.21% | 0.9120 |
| 2MT17g | 5.19 ± 0.23 | nd | 4.17% | 0.9164 |
| CRYO30g | 5.01 ± 0.14 | nd | 5.92% | 0.9158 |
| *Mentha piperita* | | | | |
| CRYOp | 13.23 ± 0.51 | 1.47 | 3.18% | 0.9100 |
| CRYO_30p | 7.55 ± 0.09 | 1.27 | 2.25% | 0.9044 |

[a] = Maximum wavelength of 322 ± 4 nm (*Hypericum*: 327 ± 2 nm); [b] = minimum wavelength of 261 ± 5 nm (*Hypericum*: 313 ± 2 nm). * Dry residue GHP specification: *Salvia* (≥1.2%); *Echinacea*, *Hypericum* (≥1.5%); *Ginkgo* (≥3.5%), *Mentha* (≥1.4%). ** Relative density GHP specification: *Echinacea*, *Hypericum* (0.890–0.915); *Ginkgo* (0.905–0.925); *Salvia*, *Mentha* (0.895–0.915). nd = not detected.

### 3.2. UV–Visible Spectrophotometry

3.2.1. Spectral Characteristic for Each MT

The MTs of *Echinacea*, *Salvia* (Figure 2A,B) and *Mentha* (Figure 3B) show an absorption spectrum characterized by the presence of two maxima at 322 ± 4 nm and 286 ± 2 nm (exception for *Echinacea*, presenting only the maximum at 322 ± 4 nm) and one minimum at 261 ± 5 nm. The MT of *Ginkgo* (Figure 2D) and *Hypericum* (Figure 2C) show similar absorption patterns to one another, with a maximum present at around 335 ± 1 nm and a minimum at around 311 ± 3 nm (less pronounced in *Ginkgo* MT).

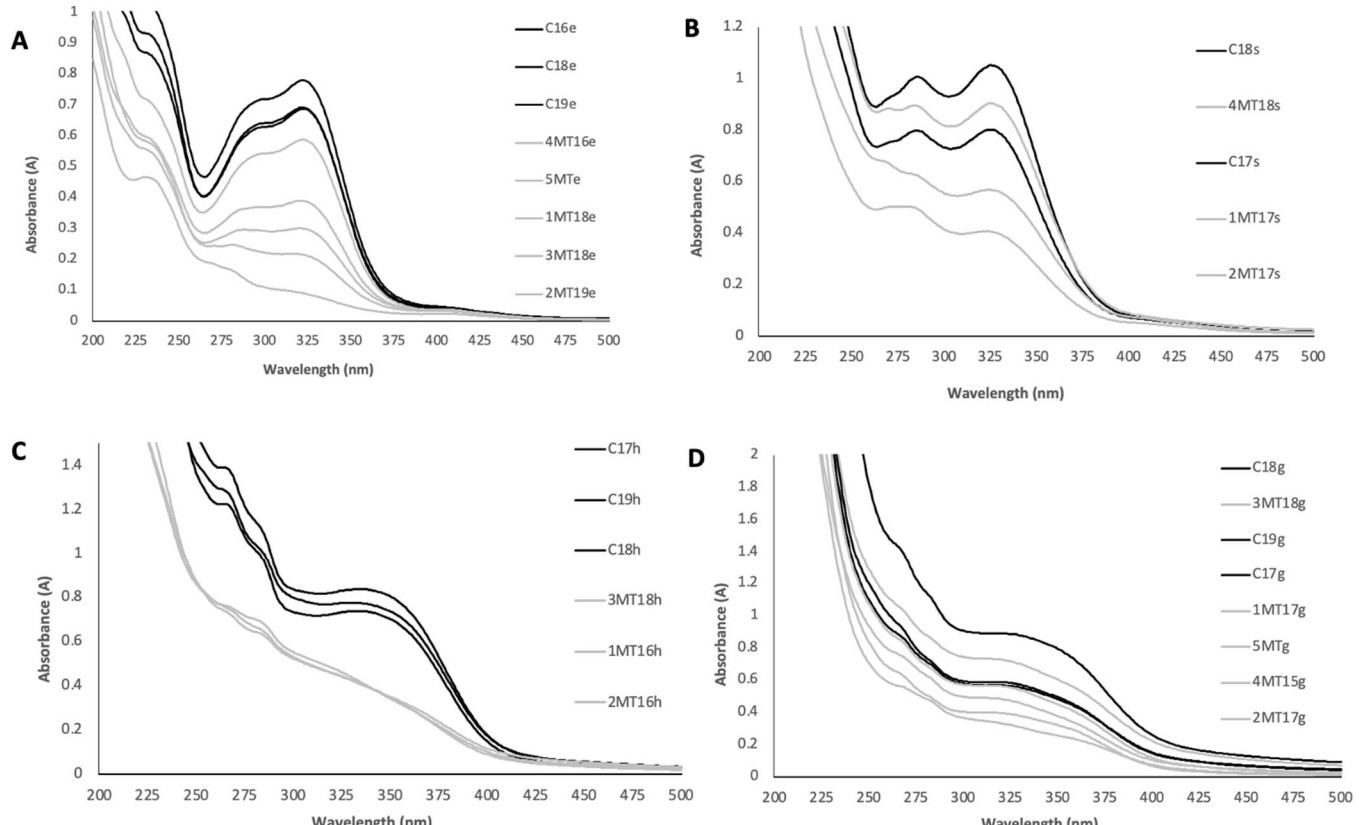

**Figure 2.** Spectrophotometric UV–Vis analysis of analyzed commercial mother tinctures of *Echinacea purpurea* (**A**), *Salvia officinalis* (**B**), *Hypericum perforatum* (**C**), and *Ginkgo biloba* (**D**). Names of extracts in the legends are ordered in descending intensity of maximum absorption.

3.2.2. Oxidation Stress Test

The results of the air exposure test (shown in Figure 3) show that the MTs obtained from the ground material of *Echinacea purpurea* and *Mentha piperita* exposed for 30 min to air oxidation exhibit an evident absorbance decrease compared to their respective CRYO oxidation free extract. The amount of dry residue was also reduced massively, as can be observed in Table 1, even if the starting material and the ratio of plant material and extractant used in the preparation of the MTs was identical. The absorbance pattern also showed a flattening of the curve, which is characterized by a diminution of the maximum/minimum ratio (see Table 1). The *Hypericum* MT shows a slight decrease in absorbance intensity and a pronounced flattening of its typical spectrum. In the case of *Ginkgo biloba* (Figure 3D), the exposure of the ground material to air oxidation for 30 min does not bring about a consistent change in the typical absorption pattern.

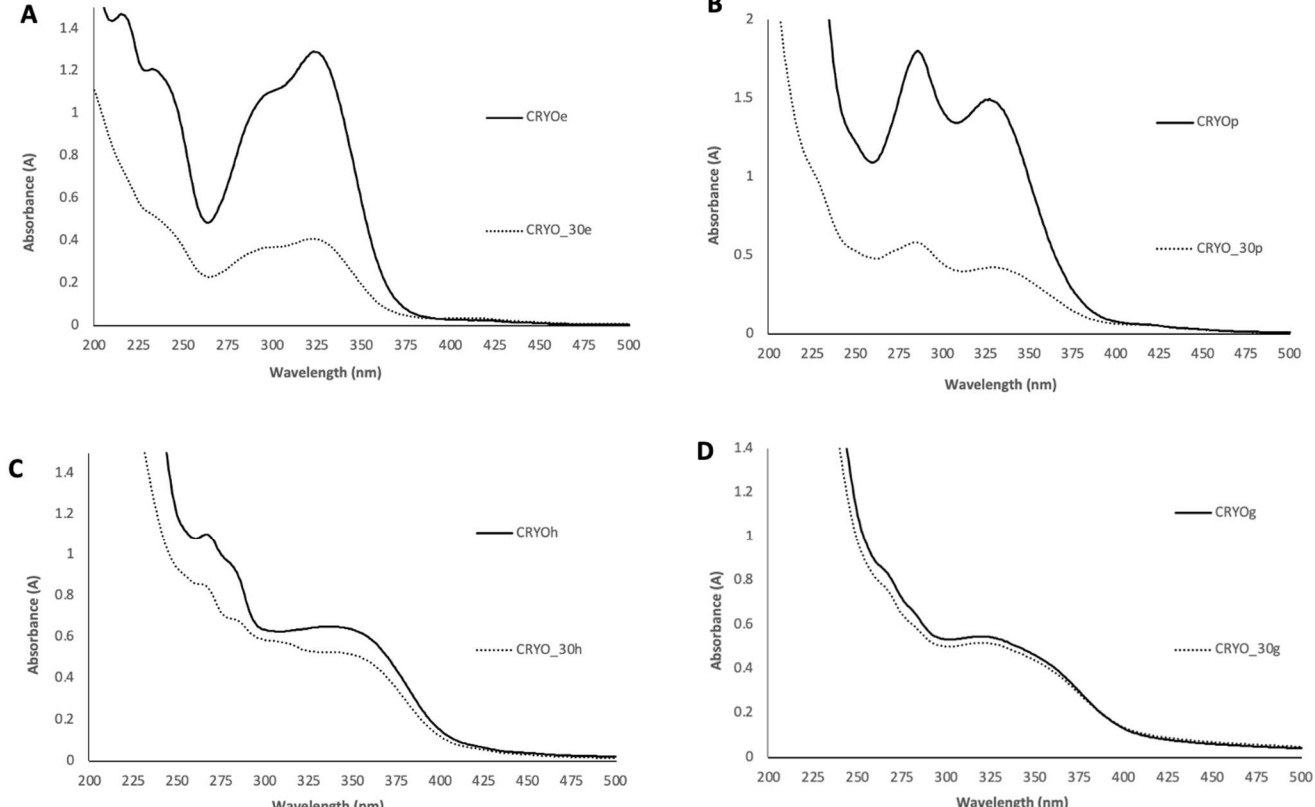

**Figure 3.** Spectrophotometric UV–Vis analysis of mother tinctures obtained in the oxidation stress test of *Echinacea purpurea* (**A**), *Mentha piperita* (**B**), *Hypericum perforatum* (**C**), and *Ginkgo biloba* (**D**) . UV–Vis spectra show the comparison between the cryoground "oxidation-free" reference (CRYO) and cryoground + 30 min air-exposed (CRYO_30) extract. Names of extracts in the legends are ordered in descending intensity of maximum absorption (between 318 and 323 nm).

### 3.2.3. Analysis of Commercial Batches

The MTs from Ceres Heilmittel AG have a generally higher absorbance compared with that of other commercial samples (Figure 2), and they show a less pronounced flattening shape, which is characterized by a higher ratio between maximum and minimum (see Table 1). An exception can be observed in the MTs of *Ginkgo* (Figure 2D), in which these differences are not evident and a high variability between samples is present.

### 3.2.4. Tyrosinase Oxidation Test

The incubation of tyrosinase with tinctures of *Echinacea purpurea* (C18e), *Mentha piperita* (C18p), *Salvia officinalis* (C18s), *Hypericum perforatum* (C17h), and *Ginkgo biloba* (C18g) resulted in a time-dependent flattening of the typical absorbance shape (Figure 4). After the addition of the tyrosinase solution, the maximum absorbance at $322 \pm 4$ nm showed a decrease, while the minimum absorbance at $261 \pm 5$ nm showed an increase (Figure 4A–C). The ratio between maximum and minimum decreased proportionally to the reaction time and appeared to occur rapidly within the first 20 min (Figure 5). In the case of the *Hypericum* (Figure 4D), a flattening of the curve was observable and expressed by a general increase in absorbance after and before the maximum at 336–334 nm. The tyrosinase oxidation assay conducted on *Ginkgo* showed no relevant difference in the absorbance shape after 180 min of reaction at the tested concentration of the enzyme (Figure 4E). In all the tinctures, except *Ginkgo*, a time-dependent increase in absorbance during oxidation was observable in the wavelength area >360 nm.

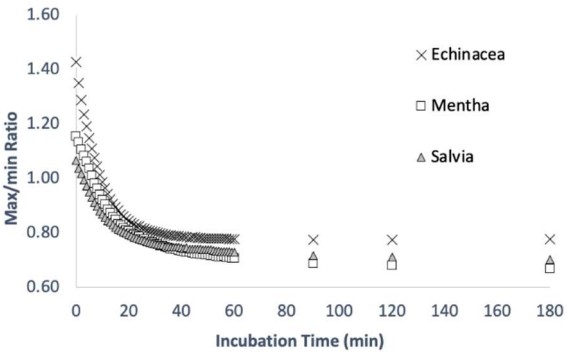

**Figure 4.** Spectrophotometric UV–Vis analysis of mother tinctures obtained in the oxidation stress test of *Echinacea purpurea* (**A**), *Mentha piperita* (**B**), *Salvia officinalis* (**C**), *Hypericum perforatum* (**D**), and *Ginkgo biloba* (**E**). UV–Vis spectra show the comparison between the cryoground "oxidation-free" reference (CRYO) and cryoground + 30 min air exposed (CRYO_30) extract. Names of extracts in the legends are ordered by descending intensity of maximum absorption (between 318 and 323 nm).

**Figure 5.** Decrease in max–min ratio over time in the tyrosinase oxidation test.

### 3.3. Potassium Permanganate: Antioxidant Assay

3.3.1. Oxidation Stress Test

In line with the general pattern of absorbance observed in the UV–Vis spectra (Figure 3), the results of the KMnO$_4$ assay (Figure 6 and Table 1) showed a significative decrease in scavenging activity (when the CRYO_30 sample is compared to the free oxidation reference CRYO), which was relative to the length of exposure to air. The results were normalized to the % dry residue values and are given as GAE per dry residue (to evaluate the antioxidant activity, avoiding the influence of the total amount of substances in the observed differences). Exposing fresh ground material to air for 30 min can lead to a decrease in the antioxidant activity (expressed as GAE) in all the tested plants in the range between 10.41% and 44.19% (see Figure 7) when compared to the respective oxidation free reference (CRYO).

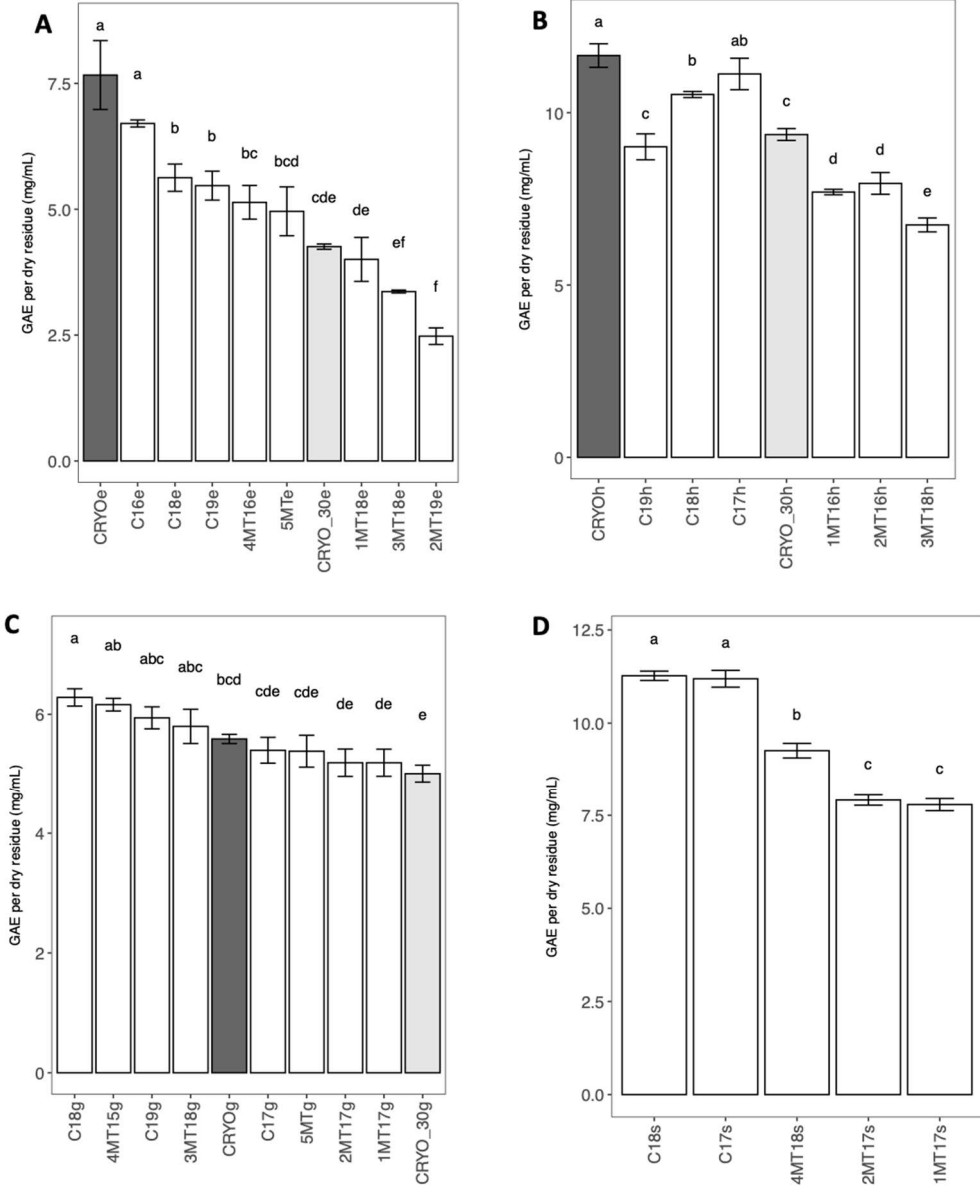

**Figure 6.** Comparison of scavenging effect obtained by means of potassium permanganate assay in the analyzed mother tinctures (*Echinacea purpurea* (**A**), *Hypericum perforatum* (**B**), *Ginkgo biloba* (**C**) and *Salvia officinalis* (**D**)) expressed as Gallic Acid equivalent (GAE) per dry residue (mg/mL). Different letters indicate the presence of significant differences between samples ($p \leq 0.05$).

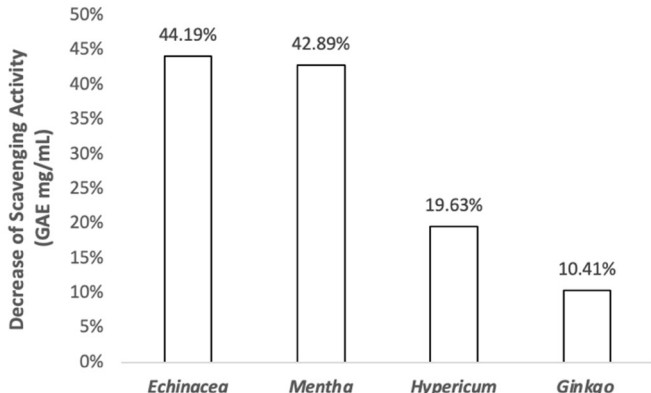

**Figure 7.** Percent decrease in scavenging affect after 30 min of oxidation obtained in the oxidation stress test for each analyzed MT.

### 3.3.2. Analysis of Commercial Batches

The comparison of commercial batches (Figure 6) showed that the MTs manufactured by Ceres Heilmittel AG showed a generally higher scavenging activity (expressed as GAE per dry residue) when compared to the other commercially available mother tinctures. The antioxidant activity of the Ceres tinctures was always between that of the CRYO oxidation free reference and the oxidized CRYO_30; instead, some of the commercial batches showed an activity lower than that of the oxidized extract CRYO_30 (Figure 6A,B). The only exception to these results was the *Ginkgo biloba* MTs (Figure 6C), in which all the commercial batches showed results between those of the CRYO and CRYO_30 extract (exception for C18g). A positive linear correlation ($R^2 > 0.7$) was found between the maximum/minimum ratio and the antioxidant activity expressed as GAE in the potassium permanganate assay (Figure 8). The correlation was present most strongly in the commercial batches of Salvia ($R^2 = 0.9116$), followed by Echinacea ($R^2 = 0.7351$). In the case of *Echinacea*, the result of the oxidation stress test was also included in the scatterplot diagram.

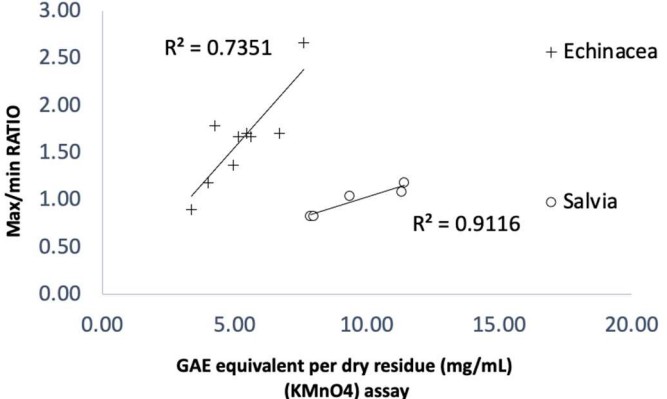

**Figure 8.** Scatterplot showing the linear correlation ($R^2 > 0.7$) present between the two variables y = max/min ratio and x = GAE equivalent per dry residue (evaluated in the KMnO$_4$ assay) recorded in the comparative analysis of commercial mother tinctures.

### 4. Discussion

The UV–Vis absorption spectra of *Mentha piperita*, and *Salvia officinalis* seem to principally be related to the presence of hydroxycinnamic acid derivatives such as rosmarinic acid, characterized by a maximum absorbance at $322 \pm 4$ nm and a minimum at 260 nm [18]. The second maximum present at $286 \pm 2$ nm could be related to the presence of other phenolic acids and flavonoids. In the work of Olennikov et al. 2010 [19], an equimolar mixture of hesperidin and rosmarinic acid shows a spectrum with a pattern corresponding to the one observed in *Mentha* in our experiment. Regarding *Salvia*, this second maximum

could be related to the presence of salvianolic acids [20,21]. The spectra of *Echinacea* correspond to the typical absorption shape of caffeic acid derivatives such as chlorogenic acid (Figure A1) and chicoric acid [22,23], known to be some of the main constituents of this plant. With regard to *Ginkgo* and *Hypericum*, their absorption spectra have some similarities and are characterized by the presence of main absorbance regions around $335 \pm 1$ nm and 260 nm. This area could contribute to the absorbance shape through the presence of certain flavonoids such as hyperoside and rutin (Figure A1), which are known to be the main flavonoids in *Hypericum* and *Ginkgo*.

When MTs are subjected to oxidation in the tyrosinase assay (Figure 4), a time-dependent flattening of the absorption pattern is observed that is characterized by a diminution of the maximum–minimum ratio (see Figure 5). Additionally, an increase in absorbance in the region >360 nm can be observed (Figure 4). Tyrosinase has the ability to oxidize the catechol moiety of phenolic compounds in benzoquinones (Figure 1), and it is known from the literature that the *o*-quinone moiety has a maximum absorbance in the area around 389 nm [13]. Different publications have shown that the oxidation of phenolic compounds such as chlorogenic acid [11], flavonoids [12], and other substances with similar structures such as trans-resveratrol [13] are characterized by an increase in absorbance in the area around 400 nm. It is for this reason that the absorbance observed in the analyzed MTs at the wavelength >360 nm could be a direct indicator of the oxidation process. This increase in absorbance can also be observed in the oxidation test shown in Figure A2, as well as the typical curve flattening (Figure 3). However, in this test we also observed a general decrease in the absorbance intensity and dry residue. *o*-quinone-oxidized products are known to be highly reactive species that can react with nucleophilic substances such as phenol and protein, producing a mixture of brown products known as melanins. The observed decrease in the dry residue could be directly due to this reaction process forming melanins. These substances can stay attached to the plant matrix and, due to their insolubility in the hydroalcoholic solvent [24], cannot pass through the MT, thus decreasing the total amount of extracted substances and the phenolic compound content.

Phenolic acids and flavonoids are mainly known for their ability to exert an important antioxidant activity in herbal remedies. If oxidation occurs during manufacturing, a decrease in the content of PCs is expected and, as a consequence, a decrease in the antioxidant activity of the extract as well. To evaluate the impact of oxidation during the grinding process, the extracts prepared in the oxidation stress test were submitted to the potassium permanganate antioxidant assay and the obtained results were normalized to those of the dry residue (thus avoiding the possible interference of the substance content in the interpretation of the data). The results (Figure 6) show a decrease in the antioxidant activity that seems to be independent of the total amount of extracted substances reflected by the dry residue (since the results were normalized) but directly related to the degree of oxidation that occurs during manufacturing.

A linear correlation seems to be present when the results of the $KMnO_4$ antioxidant assay are plotted with the values of the maximum–minimum ratio obtained from the oxidation stress test and commercial batches analyses of *Echinacea* and *Salvia* (Figure 7). The decrease in the maximum–minimum ratio seems to be related to the decrease in antioxidant activity (Figure 5) and seems to be the result of the oxidation of phenolic compounds, as shown in the UV–Vis spectra in the tyrosinase oxidation assay (Figure 4) and the oxidation stress test (Figure 3). Following these results, the observation of the UV–Vis spectra pattern and its grade of flattening could be used as a simple method, with which to obtain a first rapid evaluation of the oxidation state of an herbal mother tincture.

When the commercially available batches are compared to one another, different grades of flattening, characterized by different maximum–minimum ratios, can be observed between the MTs (Table 1 and Figure 2). The analysis of different commercially available mother tinctures showed a large amount of heterogeneity, whereby it could be shown that the tinctures of Ceres Heilmittel AG seem to exert a higher level of antioxidant activity and present a higher maximum–minimum ratio. Based on the previous hypothesis, these results

could be an indication that tinctures of Ceres are less oxidized. In the Ceres manufacturing process, plants are ground in a special mill device characterized by a closed system that is able to guarantee a reduced air contact, as well as by the presence of alcohol, which could inhibit the activity of enzymatic oxidation during the milling process [25]. This workflow could be responsible for the observed differences from other commercial batches, since a common industrial blender can expose plant material to air during grinding before extraction. However, no conclusive statement on the cause of these differences can be made, since there are too many different variables and unknown parameters, while the differences in the raw materials used between industries could be a variable leading to these differences.

Moreover, it is important to underline that not every plant has the same level of sensitivity to the oxidation process. The results show that the MTs obtained in the oxidation stress test over 30 min of air exposure showed a decrease in scavenging activity (expressed as GAE), which is more pronounced in *Echinacea* (−44.19%) and *Mentha* (−42.89%), compared to *Hypericum* (−19.63%) and *Ginkgo* (−10.41%) (Figure 7). It is interesting to note that *Ginkgo* seems to be a very stable plant during the manufacturing process when we compare with the results of the previous analysis. The compounds present in its MT seem to be very resistant to the oxidation (tyrosinase oxidation test and oxidation stress test), and, when all the commercial batches are compared, a high variability is present and no major differences can be observed between the different manufacturers.

## 5. Conclusions

The simulation of an oxidation process (oxidation and tyrosinase assay) mainly leads to: (1) a flattening of the UV–Vis spectra (characterized by a decrease in the maximum–minimum ratio; (2) a general decrease in absorbance; and (3) a decrease in antioxidant activity. A decrease in ratio and subsequent flattening were observed in the oxidation stress test as well as during the tyrosinase oxidation assay. In addition, a positive correlation was found between the decrease in the ratio of maximum to minimum and the normalized antioxidant activity. These results suggest that the interpretation of the UV–Visible spectra flattening could be used as a convenient analytical method, with which to rapidly assess the oxidation status of an MT. In conclusion, this work highlighted that, during the manufacture of a herbal mother tincture, the exposure of the plant material to air during the milling step could cause the oxidation of phenolic compounds and have a major impact on the characteristics of the final product. This impact appears to differ in relation to the plants used as the source of the herbal remedy. Some plants, such as *Echinacea purpurea* and *Mentha piperita*, seem to be strongly affected by the oxidation process, whereas plants such as *Ginkgo biloba* seem to remain very stable and not be seriously affected by the processing step. This article aims to raise awareness, especially in the industry and research on pharmaceuticals, about milled plants' sensitivity to oxidation during manufacturing processes. We believe that, by directing more attention and research to this problem, in the future, we may be able to improve the quality of remedies available on the market, thus avoiding the loss of important active substances during the production of herbal mother tinctures.

**Author Contributions:** Conceptualization, R.K., C.K., S.H. and D.B.; investigation, visualization, data curation and formal analysis, D.B.; Resources, S.H.; Supervision, S.H. and C.K.; writing—original draft preparation, D.B.; Writing—Review and Editing, S.H., C.K. and M.P. All authors provided critical feedback and helped shape the research, analysis and manuscript. All authors have read and agreed to the published version of the manuscript.

**Funding:** The research project was funded by Ceres Heilmittel AG.

**Data Availability Statement:** All of the data are contained within the article. Additional details are available from the corresponding author, C.K., upon reasonable request.

**Acknowledgments:** Special thanks to R.K. for providing the theoretical and observational foundation upon which this research project was built.

**Conflicts of Interest:** D.B., S.H., and M.P. are employees of Ceres Heilmittel AG. C.K. is the CEO of Ceres Heilmittel AG. R.K. is the founder of Ceres Heilmittel AG.

## Appendix A. Additional Results on UV–Visible Spectrophotometry

This appendix is an integration of the discussion part of paragraph 4, consisting in two additional UV-Vis spectra. Standard solutions of chlorogenic acid, rutin and hyperoside (Figure A1) were analyzed and compared with the absorbance spectra of the analyzed mother tinctures (Figure 2). Figure A2 shows the absorbance region between 350 and 500 nm of *Echinacea* mother tinctures obtained in the oxidation stress test. The increase in absorbance in this region could be related to the presence of *o*-quinones derived from the oxidation process of phenolic compounds [13].

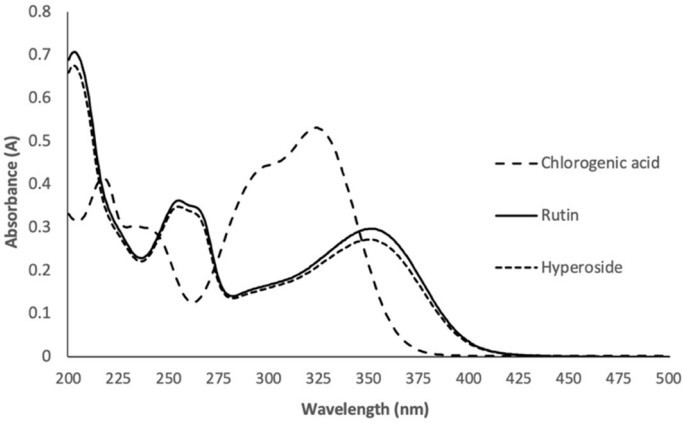

**Figure A1.** Standard solution of chlorogenic acid, rutin, and hyperoside (1 mg/mL methanol) diluted 1:100 *v/v* with distilled water and analyzed between 200 and 500 nm.

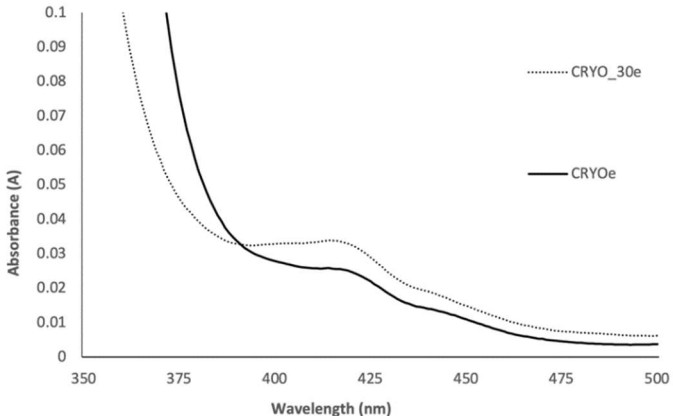

**Figure A2.** Spectrophotometric UV–Vis analysis of mother tinctures obtained in the oxidation stress test of *Echinacea purpurea*. UV–Vis spectra show the comparison between the cryoground "oxidation-free" reference (CRYO) and cryoground + 30 min air-exposed (CRYO_30) extract. The figure is a scale zoom of Figure 3A to underline the increase in absorbance in the wavelength region between 350 and 500 nm typical of the *o*-quinone moiety.

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
