# Peer review of "Oxidation during Fresh Plant Processing: A Race against Time"

_processes, doi:10.3390/pr10071335_

Round 1

Reviewer 1 Report

Observations regarding the article entitled: Oxidation during fresh plant processing: A race against time” authors Didier Barmaverain et al.,

1.      The meaning of the term GHP must be specified.

2.       "Standard GHP method for MT control", paragraph 2.4., line 108, page 8, must be referenced

3.      Bibliographic reference for the method: "2.7. Tyrosinase oxidation assay", line 143, page 4

4.      How are the results for scavenging effect as GAE per dry residue (mg/mL) expressed?  How were the values in Table 1 (page 5) for scavenging effect expressed as GAE calculated?

Author Response

Dear reviewer,

Thank you very much for your review and helpful comments that can improve the quality of our work. The manuscript has been updated by incorporating the feedback you proposed. You will find answers to your questions below.

  1. The meaning of GHP has been introduced in paragraph 2.3., line 105. The paragraph has been corrected as follows: All MTs were prepared according to German Homeopathic Pharmacopoeia Method 3a (GHP) [14].

A new reference regarding production method 3a has been introduced (Stephen Benyunes. Methods for the production of homeopathic medicines. In: German Homoeopathic Pharmacopoeia. medpharm GmbH Scientific Publishers; 2003. p. 23.).

  1. Reference to the GHP standard method has been introduced in paragraph 2.4., line 108, as requested.

Europäisches Arzneibuch. 9th ed. Vol. 9. Deutscher Apotheker Verlag, Govi-Verlag - Pharmazeutischer Verlag; 2017. 33, 399 p.

  1. The reference for method 2.7. Tyrosinase oxidation assay," has been inserted in line 144, paragraph 2.7, by including the following new sentence: "The tyrosinase oxidation assay was conducted by adapting the enzyme assay described by Tono et al. 1987 [11] and Oda et al. 1989 [16], with a few modifications."

       4. The scavenging effect results as GAE per dry residue (mg/mL) were expressed by relating the % scavenging effect obtained from the sample to the % scavenging effect obtained in the calibration curve using an aqueous dilution of gallic acid in the following concentration range (0-3-6-12-24-48 µg/mL).

The % scavenging effect was calculated as follows: ((Abs Blank - Abs Sample)/Abs Blank)*100

GAE was calculated by applying the calibration curve equation (y = 1.3995x + 1.7649) to the % scavenging effect as follows:

GAE (x) = (% sample scavenging effect (y) - 1.7649)/1.3995

GAE were further normalized by dry residue as follows:

GAE normalized by dry residue = GAE / % sample dry residue

The final values obtained in µg/mL refer to the concentration present in the tube at the time of the assay. The GAE contents present in 1 mL of undiluted tincture were calculated by multiplying the normalized GAE per dry residue with the *dilution factor (DF) 1000. Final values were converted from µg/mL to mg/mL by dividing by 1000.

*Dilution factor: The tinctures were diluted for the assay 1:25 -> DF = 25 and then were diluted 1:4 during the assay (1 mL of 1:25 diluted tincture + 3 mL of KMnO4 working solution -> DF = 4). The µg of GAE present in 1 mL of undiluted tincture was calculated by multiplying the GAE normalized by dry residue with the dilution factor 1000.

I remain available for future clarifications.

Sincerely.

Christoph Kalbermatten

Reviewer 2 Report

El trabajo tiene resultados novedosos relacionados con el control de calidad de productos fitoterapéuticos. Consideré algunas observaciones en la sección de metodología y discusión que se explicarán para una mejor compresión del manuscrito:

-No se determinó el contenido total de PC con los métodos de folin-ciocalteu; ¿Por qué no se determinó? La determinación del total de PC proporcionaría más información para el presente estudio.

- Sería importante incluir en la determinación del método de ensayo antioxidante del permanganato de potasio su precisión, linealidad y sensibilidad del método UV-Vis.

- ¿Por qué no se utilizaron compuestos de referencia como estándares fenólicos para determinaciones de antioxidantes y ensayos de oxidaciones?

-Incluir en las referencias el método GHP 3a

- Por qué hay diferencias en las concentraciones de solución de tirosinasa (8,3 U/mL para Echinacea, Salvia y Mentha y 16,2 U/mL para 146 Hypericum y Ginkgo)

-No registrados en la Tabla No 1, los valores de CRYO y CRYO30 para Salvia y Ginkgo biloba.

-Los autores deben explicarse mayormente la afirmación de las líneas 338 a 391: Los resultados (Figura 6) muestran una disminución en la actividad antioxidante que parece ser independiente de la cantidad total de sustancias (ya que los resultados se normalizaron) pero directamente relacionada al grado de oxidación que se produce durante la fabricación . ¿Esta afirmación tiene relación con la cantidad de materia vegetal utilizada en el análisis de antioxidantes y oxidantes? la oración cantidad total de sustancias puede causar confusión y relacionarse con el contenido de las PC.

Author Response

Dear reviewer,

Thank you very much for your review and helpful comments that can improve the quality of our work. The manuscript has been updated by incorporating the feedback you proposed where possible with the data we have available at the moment. If additional experiments are required for the improvement of the manuscript and its publication, we remain willing to perform additional experiments in the future to be integrated later in the manuscript. You will find answers to your questions below.

1. 

The Folin Ciocalteau test was only preliminarily performed in the analysis of an Echinacea sample. A similar relative trend to the results of the permanganate test was observed. We agree that the inclusion of further experiments with the FOLIN-Ciocalteau reagent could provide additional information in our study, as you suggested. If necessary for publication, we could add an additional experiment on total polyphenol content by FOLIN-CIocalteau and update the manuscript.

2. 

The linearity of the method was established by preparing a calibration curve with increasing concentration of gallic acid (0-3-6-12-24-48 µg/mL). The linear regression curve is as follows: r2 = 0.995; y = 1.3955x + 1.7649.

The regression line is described in line 165. Should the graph be included in the text?

Regarding sensitivity, the limit of detection (LOD) we’ve calculated corresponds to 0.02 µg/mL and the limit of quantification (LOQ) to 0.05 µg/mL. (These values are relative to GAE before normalization, which are not presented in the manuscript, but can be calculated by multiplying GAE values by the respective % dry residue.)

Precision was not evaluated, but we refer to the cited work of Amponsah et. al 2016. If necessary to improve the manuscript for publication, an additional experiment could be performed to evaluate precision and integrate the results (including also the LOD and LOQ together) in a dedicated part of the manuscript.

3. 

As for the reference compound, the gallic acid standard was used to perform the potassium permanganate antioxidant test and establish the calibration curve to evaluate the gallic acid equivalent of each sample analyzed. Because of the similarity of the absorption spectra of Echinacea with the absorption spectrum of chlorogenic acid (shown in Supplementary Figure S1 in the manuscript), the tyrosinase assay was conducted only on the mother tincture samples, and the assumption of conducting the test on the pure reference compound was not considered. If necessary to improve the manuscript, an additional experiment including oxidation by tyrosinase of the selected standard could be conducted and integrated into the manuscript later.

4.

The reference of the method GHP was introduced at paragraph 2.3., line 105. The paragraph was corrected as follow: All MTs were prepared according to the German Homeopathic Pharmacopeia (GHP) method 3a [14].

A new reference related to the manufacturing method 3a was introduced (Stephen Benyunes. Methods for the Production of Homoeopathic Medicinal Product. In: German Homoeopathic Pharmacopoeia. medpharm GmbH Scientific Publishers; 2003. p. 23.).

5.

These differences are due to the fact that the first concentration (8.3 U/ml) was initially tested on Echinacea and being optimal was successfully applied to the other tinctures. Nevertheless, during the test on Ginkgo (the last tincture tested) no flattening of the curve was observed. For this reason, we decided to dilute the enzyme concentration less, bringing it to 1:50 (16.2 U/mL) instead of 1:100 (8.3 U/mL), and to test oxidation on the Ginkgo tincture. Given the similar absorption pattern to Hypericum, this concentration was also tested on this tincture. However, this concentration was unable to oxidize the Ginkgo tincture, while the Hypericum tincture showed a slight increase in flattening. We decided to publish these different enzyme concentrations related to Ginkgo and Hypericum to emphasize the oxidative stability of Ginkgo, which was also observed in the oxidative stress test performed with liquid nitrogen.

6.

The oxidative stress test including the results of CRYO and CRYO30 extracts was not performed for Salvia (as described in the manuscript in lines 134-135). This is because all plants included in the oxidative stress test were harvested by the company's production department and sampled in the R&D laboratory on the same day in Kesswil, Switzerland. Salvia officinalis is processed in Nax (the second production department) instead of Kesswil, and due to its unavailability at the main site in Kesswil and the problem of transportation distance, the test on Salvia officinalis was not performed.

In the case of Ginkgo, CRYO 30 values were included in the table in the updated manuscript.

7.

The statement "total amount of substances" refers to the % dry residue of each tincture analyzed and not to the total polyphenol content. The % dry residue of a mother tincture is an indicator of the total amount of substances extracted in the hydroalcoholic solvent from the plant matrix after maceration. A higher content of antioxidant substances (polyphenols and non-polyphenols) could react with permanganate giving higher antioxidant activity. Normalizing the results obtained with the dry residue was one way to obtain a result of antioxidant activity that is related to the actual oxidation state of the molecules in the tinctures (qualitative aspect). In this way, the results could be independent of the total amount of extracted substances (quantitative aspect) and directly reflect the oxidative state of the mother tinctures in the results.

The total amount of substances and its relationship to the dry residue are first mentioned on line 178 of page 5 as follows: "The percentage of dry residue of the MTs analyzed, representing the total amount of substances extracted and their respective relative density, is shown in Table 1."

To avoid confusion, the sentence in line 387-390 has been modified as follows in the manuscript: "The results (Figure 6) show a decrease in antioxidant activity that appears to be independent of the total amount of extracted substances reflected by the dry residue (since the results were normalized), but directly related to the degree of oxidation occurring during production". If additional adaptation are required to a better clarification, we’re at disposition to implement the discussion part on the total amount of extracted substances expressed by the dry residue, to avoid confusion to the readers.

I remain available for future clarifications.

Sincerely.

Christoph Kalbermatten